Review of feature selection approaches based on grouping of features

Kuzudisli Cihan cihan.kuzudisli@hku.edu.tr 1 2
Bakir-Gungor Burcu 3
Bulut Nurten 3
Qaqish Bahjat 4
Yousef Malik malik.yousef@gmail.com 5 6
1 Department of Computer Engineering, Hasan Kalyoncu University , Gaziantep , Turkey
2 Department of Electrical and Computer Engineering, Abdullah Gul University , Kayseri , Turkey
3 Department of Computer Engineering, Abdullah Gul University , Kayseri , Turkey
4 Department of Biostatistics, University of North Carolina at Chapel Hill , North Carolina , Chapel Hill , United States of America
5 Department of Information Systems, Zefat Academic College , Zefat , Israel
6 Galilee Digital Health Research Center, Zefat Academic College , Zefat , Israel
Singh Reema
Electronic publication date: 2023 Jul 17
Publication date: 2023
Volume: 11
Electronic Location ID: e15666
Received 2022 Nov 14; Accepted 2023 Jun 8
Copyright: ©2023 Kuzudisli et al.
Copyright year: 2023
Copyright holder: Kuzudisli et al.
License: This is an open access article distributed under the terms of the Creative Commons Attribution License, which permits unrestricted use, distribution, reproduction and adaptation in any medium and for any purpose provided that it is properly attributed. For attribution, the original author(s), title, publication source (PeerJ) and either DOI or URL of the article must be cited.
License URL: https://creativecommons.org/licenses/by/4.0/

Keywords: Feature selection, Feature grouping, Supervised, Unsupervised, Integrative

Funding: Zefat Academic College Abdullah Gul University Support Foundation (AGUV) This work has been supported by the Zefat Academic College. Burcu Bakir-Gungor’s work has been supported by the Abdullah Gul University Support Foundation (AGUV). The funders had no role in study design, data collection and analysis, decision to publish, or preparation of the manuscript.

==============================
With the rapid development in technology, large amounts of high-dimensional data have been generated. This high dimensionality including redundancy and irrelevancy poses a great challenge in data analysis and decision making. Feature selection (FS) is an effective way to reduce dimensionality by eliminating redundant and irrelevant data. Most traditional FS approaches score and rank each feature individually; and then perform FS either by eliminating lower ranked features or by retaining highly-ranked features. In this review, we discuss an emerging approach to FS that is based on initially grouping features, then scoring groups of features rather than scoring individual features. Despite the presence of reviews on clustering and FS algorithms, to the best of our knowledge, this is the first review focusing on FS techniques based on grouping. The typical idea behind FS through grouping is to generate groups of similar features with dissimilarity between groups, then select representative features from each cluster. Approaches under supervised, unsupervised, semi supervised and integrative frameworks are explored. The comparison of experimental results indicates the effectiveness of sequential, optimization-based (i.e., fuzzy or evolutionary), hybrid and multi-method approaches. When it comes to biological data, the involvement of external biological sources can improve analysis results. We hope this work’s findings can guide effective design of new FS approaches using feature grouping.

Introduction

In the current digital era, the data produced by many applications in fields such as image processing, pattern recognition, machine learning and network communication grow exponentially in both dimension and size. Due to this high-dimensionality, the search space is widening and extraction of valuable knowledge from the data becomes a challenging task (Abdulwahab, Ajitha & Saif, 2022; Venkatesh & Anuradha, 2019). Also, utilizing all features in a dataset is unlikely to develop a predictive model with high accuracy. The existence of irrelevant and redundant features may weaken the generalizability of the model and decrease the overall precision of a classifier (Jovic, Brkic & Bogunovic, 2015). Hence, reducing the number of input variables is highly desired as it lowers the computational cost of model construction and allows improving model performance. As such, feature selection (FS) becomes an inevitable step for domain experts and data analysts.

FS is the process of selecting the minimally sized feature subset from the original set that is optimal for the target concept. It plays a crucial role in removing irrelevant and redundant features while keeping relevant and non-redundant ones (Md Mehedi, Mollick & Yasmin, 2022). Irrelevant features do not alter the target concept in any way and redundant features do not contribute to the target concept (John, Kohavi & Pfleger, 1994). These features may contain a considerable amount of noise which can be misleading, resulting in significant computational overhead and poor predictor performance. Contrary to other dimensionality reduction techniques, FS preserves the data semantics as it does not distort the original feature representation and hence provides straightforward data interpretation for data scientists. Additionally, reduction in dimension by FS prevents overfitting that can lead to undesired validation results.

Although various FS techniques have been developed, traditional approaches to FS neglect structures of features during the selection process. Another issue is that retention and elimination of features on an individual basis ignores dependence among them. Because of these reasons, correlation between features may not be detected efficiently resulting in irrelevant or redundant features in the final subset. Some studies grouped samples (i.e., observations) for improving classification performance but these studies were not concerned with feature reduction at all (Wang, Wu & Zhang, 2005; Maokuan, Yusheng & Honghai, 2004).

On the other hand, FS based on grouping is an effective technique for reducing feature redundancy and enhancing classifier learning. By grouping the features, the search space is reduced substantially. Moreover, it can reduce estimator variance (Shen & Huang, 2010), improve stability, and reinforce generalization capability of the model. Although there are reviews of clustering methods (Mittal et al., 2019) and of FS techniques (Venkatesh & Anuradha, 2019; Chandrashekar & Sahin, 2014), to the best of our knowledge, this is the first article reviewing the literature on approaches to FS based on grouping. In this procedure, the process of grouping features into clusters is generally performed as the initial step, aiming to have maximal intra-class similarity (i.e., similarity in between the objects of the same cluster) and minimal inter-class similarity (i.e., objects in a cluster are more similar to those in another one) between features. These feature groups can be created by K-means, fuzzy c-mean (FCM), hierarchical clustering, graph theory and other methods (Dai et al., 2022; Ravishanker et al., 2022; Rashid et al., 2020; AbdAllah et al., 2017). After cluster formation, features within each cluster are scored and selected using various techniques or metrics.

The remainder of this article is organized as follows: we will give a concise overview of different FS methods in ‘Survey Methodology’. In ‘Feature Selection Approaches’, we will present different works carried out in FS using feature grouping following the summary of traditional approaches. Then, in ‘ Feature Grouping with Recursive Cluster Elimination’, we will review different studies which benefited from recursive cluster elimination based on support vector machine (SVM-RCE) (Yousef et al., 2007; Yousef, Jabeer & Bakir-Gungor, 2021; Yousef et al., 2021a). Next, in ‘Grouping Features with Biological Domain Knowledge’, we will address FS techniques involving both feature grouping and incorporating domain knowledge. We discuss the advantages and disadvantages of the presented methods in ‘Discussion’. Lastly, in ‘Conclusions’, we conclude our review with further discussions and future directions.

Rationale behind the review and intended audience

Nowadays, the advancements in different technologies resulted in the generation of high dimensional data in many different fields, which makes data analysis a challenging issue. Existence of irrelevant and redundant features makes it hard to infer meaningful conclusions from data, degrades model performance and leads to computational overhead. Especially in the field of molecular biology, the advancements in high throughput technologies have induced the emergence of a wealth of -omics data produced by different studies, such as genomics, transcriptomics, epigenomics, proteomics, meta-genomics, meta-transcriptomics, meta-proteomics, metabolomics, etc (Md Farid, Nowe & Manderick, 2016). For instance, high-dimensional RNA-sequencing data can be used for cancer subtype identification in order to ease cancer diagnosis and discover effective treatments. However, only a subset of features (i.e., mRNAs) carries information associated with the cancer subtype. Furthermore, this kind of biological data often involves redundant and irrelevant features which can mislead the learning procedure in modeling and can cause overfitting. As another example, in metagenomics studies the number of features (i.e., taxa) is much higher than the number of samples. This phenomenon is known as the curse of dimensionality. In this respect, some metagenomics studies focus on the FS process rather than focusing on classification (Bakir-Gungor et al., 2022). Hence, FS has become a real prerequisite in the biological domain (Li et al., 2022; Bhadra et al., 2022; Manikandan & Abirami, 2021; Remeseiro & Bolon-Canedo, 2019). Due to these reasons, FS became an indispensable preprocessing step in different fields dealing with high dimensional data. Traditional approaches evaluate features without considering the correlation among them, and also this evaluation is performed on an individual basis. Furthermore, these methods generally fail to scale on a large space.

On the other hand, FS based on feature groping is a powerful approach due to the following reasons: (i) it enables the discovery of correlations among features, (ii) search space is significantly diminished, (iii) it relieves computational burden. Although some grouping-based FS methods are proposed in the literature, to the best of our knowledge, none of the existing articles evaluate these existing approaches in detail as a review. For these reasons, compared to current literature, we believe that this review will be more guiding and suggestive for those learning the above-mentioned methods, for those working to derive such methods, and for those who want to apply this approach into their data analysis.

Survey methodology

Our main focus in this review is to examine FS approaches via grouping. In this context, we reviewed Web of Science, Scopus, and Google Scholar on January 10, 2022 using the following query: “feature clustering” OR “feature grouping” OR “clustering based feature selection” OR “grouping based feature selection” OR “cluster based feature selection” OR “group based feature selection”. We excluded those studies grouping samples (i.e., observations) or features as the final outcome and those concerned with feature extraction. We particularly focused on grouping of features as the preprocessing step followed by extraction of a reduced subset of features by a certain procedure which is subsequently input into a classification or clustering process for validation. Other articles for context were added while writing the review. Studies of this paradigm under an unsupervised setting are on a limited scale compared to the supervised setting, due to lack of labels in the former. Even though it is not known clearly, we think that inclusion of this approach may have emerged in late 90s. Recently, interest in this concept has grown rapidly in different forms as we point out in the following sections of this review. In fact, selection of significant features by removing irrelevant or redundant ones is just one aspect; ranking of these features in terms of being informative or having discriminative power, and stability of them for different models are other issues that are taken into consideration. Here, we examined different studies that are identified in literature mining, categorized them, and presented readers a versatile work in which we aimed at providing a robust basis on the topic.

Basics of Feature Selection

In this section, we present basic concepts in FS. According to their interaction with the classification model, FS techniques can be classified into filter, wrapper, and embedded techniques (Kohavi & John, 1997). Later in the literature, hybrid and ensemble techniques have emerged as variants of them. Hybrid approach combines two different methods to utilize the advantages of both approaches, where the common combination is filter and wrapper methods. Ensemble technique integrates an ensemble of feature subsets and then yields the result from the ensemble. The overview of the three main types of methods is shown in Fig. 1.

Figure 1 Three basic types of FS methods.

(A) Filter. (B) Wrapper. (C) Embedded.

Filter method

Filter type methods select features by assessing intrinsic properties of data based on statistical measures instead of cross-validation performance. They are easily scalable to high-dimensional datasets, independent of the learning algorithm; they are simple and computationally fast; and they are resistant to overfitting. In this method, each feature is assigned a score determined by the selected statistical method. Afterwards, all features are ranked in descending order and those with low scores are removed using a threshold value. The remaining features comprise the feature subset and are then fed into the classification model. Consequently, FS is carried out once and then various classifiers can be employed. Disadvantages of this technique are (i) features are selected irrespective of the classifier, and (ii) feature dependencies are ignored. Some common statistical measures used in this technique are information gain (IG), Pearson’s correlation (PS), Chi square ( χ2), mutual information (MI), and symmetrical uncertainty (SU).

Information gain

Information gain (IG) (Hall & Smith, 1998) is an entropy-based FS method and used to measure how much information a feature carries about the target variable. IG of a feature X in a data group D with n class labels, IG(X), is calculated using (1) IGX=ED−∑i=1nDiDEDi

where E(D) denotes the general entropy belonging to class labels, DiD is the ratio of number of occurrences of each value on feature X, and E(Di) specifies the entropy of ith feature value calculated by splitting dataset D based on feature X. Entropy is a measurement of unpredictability or impurity of a data distribution and defined as: (2) ED=−∑i=1npilog2pi

where p(i) is the probability of class i in the data group D for n class labels. A feature is relevant to target variable if it has a high information gain. The way the features are selected is in a univariate way (i.e., features are selected independently), therefore, redundant features cannot be eliminated in this technique.

Pearson’s correlation

Pearson’s correlation is a measure of the dependency (or similarity) of two variables and used for finding the relationship between continuous features and the target feature (Press et al., 2007; Nettleton, 2014). It produces the correlation coefficient r ranging between −1 to 1, where 1 shows a strong correlation and −1 means a total negative correlation. So, 0 value implies no correlation between the features. A positive correlation states that if one variable increases, so does the other variable, whereas a negative correlation implies that while one variable raises, another one decreases. This method can also be used to measure correlation between pairs of features. In this way redundant features can be identified. Pearson’s correlation coefficient r can be found for feature X with values x and classes Y with values y where X, Y are random variables by the following equation: (3) r=∑x−x ¯y−y ¯∑x−x ¯2 ∑y−y ¯2

where x ¯ and y ¯ are means of x and y, respectively. Note that Pearson’s correlation is mainly covariance of two variables divided by product of their standard deviations.

Chi square

Chi square (χ2) (Liu & Setiono, 1995) is a statistical method to test the independence of two events. It is a measurement of the degree of association between two categorical values. It measures the deviation from the expected frequency assuming the feature event is independent of the class label. This assumption is tested for a given feature with n class and m different feature values by the formula (4) χ2= ∑i=1m ∑j=1nOij−Eij2Eij

where Oij is the observed (i.e., actual) value and Eij refers to the expected value suggested by the null hypothesis. Eij is calculated as (5) Eij=O∗jOi∗O

where O∗j means the number of samples in class m, and Oi∗ indicates the number of samples with the ith feature value for the feature under study. The higher value of χ2 shows rejection to the null hypothesis, namely, higher dependency between the feature and the class label.

Mutual information

Mutual information (MI) (Cover & Thomas, 2005) is another statistical method used to assess the mutual dependence between the two variables. MI quantifies the amount of information that one random variable includes in the other random variable. MI between two continuous random variables X and Y with their joint probability functions p(x, y), and their marginal probability density functions p(x) and p(y), respectively is given by (6) IX;Y=∬px,ylogpx,ypxpydxdy.

For discrete random variables, the double integral is substituted by a summation as (7) IX;Y= ∑x∈X ∑y∈Ypx,ylogpx,ypxpy.

We can also define the conditional mutual information (CMI) of two random variables X and Y given a third variable Z as (8) IX;Y|Z=∭px,y,zlogpx,y|zpx,zpy|zdxdydz

It can be interpreted as the amount of information X includes in Y which is not shared by Z.

Symmetrical uncertainty

This is one of the techniques that are used to measure redundancy between two random variables (Witten, Frank & Hall, 2011). It is obtained by normalizing MI to the entropies of two variables and limiting it to the range of [0,1]. It’s able to circumvent inherent bias of MI toward features with a wide range of different values. Symmetrical uncertainty (SU) is defined as (9) SUX,Y=2MIX,YHX+HY

where H(X) and H(Y) are entropy of variable X and Y, respectively. A value 1 between a pair of features indicates that knowledge of feature value can fully predict the values of other and 0 value shows that X and Y are not correlated.

Based on SU, C-Relevance between a feature and a target variable C, and F-Correlation between feature pair can be defined as follows (Song, Ni & Wang, 2013):

C-Relevance: SU between feature Fi ∈F and target variable C, denoted by SUi,c.

F-Correlation: SU between any feature pair Fi and Fj (i ≠ j), denoted by SUi,j.

Wrapper method

In this methodology, a search strategy for possible subsets of features is defined, and the learning algorithm is trained using these subsets in an iterative manner. Unlike filter methods, wrapper methods are in interaction with the classifier, however, the evaluation of feature subsets is obtained using a specific classification model which makes this method specific to a learning model. Several possible combinations of features are evaluated in the model by wrapping the search algorithm around it (Visalakshi & Radha, 2014). This method provides suboptimal feature subsets for training the model since evaluating all possible subsets is computationally not practical, and generally gives better predictive accuracy than filter methods but is computationally intensive due to searching overhead and learner dependence.

The search for generating subsets can be performed with schemes such as forward selection, backward elimination, stepwise selection or a heuristic search (Liu & Motoda, 1998). Forward selection is a repetitive technique where no feature is considered at the onset. Initially, the feature with the best performance is added. Then another more significant feature giving the best performance together with the previously added feature is selected. This process proceeds until the inclusion of a new feature does not improve the classifier performance. In backward elimination, the algorithm starts with all the features available and discards the most insignificant feature from the model recursively. This elimination process is repeated until removal of features does not enhance the performance of the model. For stepwise selection, this technique is a combination of both forward selection and backward elimination. It starts with an empty set and the most significant feature is added at each iteration. While adding a new feature, previously selected features are removed if any of them has become insignificant. Heuristic search is concerned with optimization and aims at optimizing the objective function in evaluation of different subsets (Liu & Yu, 2005).

Support Vector Machines with Recursive Feature Elimination (SVM-RFE) (Guyon et al., 2002) is a popular example of wrapper methods. The idea is mainly to train the classifier by the given data and assign a rank by SVM for each feature as its weight. Then, features with the smallest weights are removed by a specific rate determined by the user. This procedure is repeated until reaching a predefined number of features.

Embedded method

This method includes advantages of filter and wrapper methods and performs FS and model construction at the same time. Just like wrapper techniques, they are specific to a learning model but they have less computational complexity than wrapper methods (Li et al., 2018). One technique of this type of FS is regularization that adds a penalty to the coefficients to overcome overfitting in the model. As an example, Lasso (Tibshirani, 1996) is an embedded method that uses L1 norm of the coefficient of a linear classifier w and penalty term (φ) is defined as (10) φw= ∑i=1k|wi|

and (11) w ˆ= minwcw,X+αφ

where c(.) is the objective function for classification, φ is a regularization term, k is the number of features, α is the regularization parameter controlling the trade-off between the objective function and the penalty. These coefficients may even be reduced to 0 for features that do not contribute to the model. Features with non-zero coefficients are retained and those with low or zero coefficient are excluded (Tang, Alelyani & Liu, 2014). Another technique to integrate FS in model creation is decision trees. These tree-based methods are non-parametric models that consider features as nodes. Tree-based strategies used by random forests accumulate various numbers of decision trees and rank the nodes (i.e., features) by decrease in the impurity (e.g., Gini impurity) over all the trees, e.g., classification and regression tree (CART) (Breiman et al., 2017).

Feature Selection Approaches

Broadly speaking, FS algorithms conducted in many studies can be categorized into the following two classes: (i) traditional FS, (ii) FS based on grouping. Traditional approaches generally consider all features contingent on “singularity” during the selection process. To put it another way, they comprise inclusion or elimination of features based on some statistical measures or classifying capacity at a singular level. On the other hand, grouping-based methods detect relevant features by grouping them into clusters; and then remove redundant ones which lead to reduced search space.

Traditional feature selection

Different FS methods exist in abundance in the literature, including filters based on distinct criteria such as dependency, information, distance and consistency (Dash & Liu, 1997), and wrapper and embedded methods employing different induction algorithms. Due to their simplicity, filter methods are often preferable in the context of high dimensional data; the absence of necessity for a search route and the interaction with a classifier makes them computationally efficient and practically feasible in applications. A comparative study on various filtering methods including mixture model, regression modeling and t-test was presented in Pan (2002) where the authors outlined similar and dissimilar aspects of these methods. The authors noted that all the three methods employ two-sample t-test or its variation; but these methods vary in different significance levels and the number of detected features. Lazar et al. (2012) also reviewed filter type FS algorithms used in gene expression data analysis and presented them as a top-bottom strategy in a taxonomy.

Wrapper methods carry the computational burden since they require navigation in the search domain and and since they interact with the predictor. However, they provide better accuracy than filter approaches due to their interaction with the learning algorithm. Talavera (2005) compared filter and wrapper approaches in clustering. They confirm the superiority of wrappers along with some of their problems and they suggest filter techniques as an alternative approach due to their computational efficiency. A recent study by ElAboudi & Benhlima (2016) overviewed existing wrapper techniques and evaluated the pros and cons of them. Embedded methods, like wrapper techniques, possess computational complexity when it comes to high-dimensional data. They are more efficient than wrappers and have less complexity. Applications of this approach in the bioinformatics domain have been reviewed in Ma & Huang (2008).

Hybrid methods combine two methods such as filter and wrapper to take advantage of both methods in order to increase efficiency and performance. Ensemble methods integrate different methods for FS, classification or both. In this approach, multiple feature selectors, induction algorithms, different subsets may be included according to the design scheme. A detailed discussion on hybrid methods and a good review on ensemble FS techniques can be found in Asir, Appavu & Jebamalar (2016) and Bolón-Canedo & Alonso-Betanzos (2019), respectively. In some studies, FS methods are divided into these five categories (Ang et al., 2016).

Traditional FS approaches have several shortcomings. For instance, filter methods evaluate the significance of each feature individually without considering the relationships and interactions between the features. Wrapper methods can provide the optimal feature subset but their complexity makes them imperfect, they are not preferable especially in combinatorial techniques such as in ensemble methods. In addition, they are not applicable to data with small number of samples due to overfitting. Embedded methods, like wrappers, are specific to the model hence may give a different feature subset for the same dataset. The main drawback behind such methods is their inability to remove redundant features and retain informative features efficiently (Khaire & Dhanalakshmi, 2022; Kamalov, Thabtah & Leung, 2022).

Feature selection through feature grouping

In this section, we will categorize FS approaches based on feature grouping under supervised, unsupervised and semi-supervised context. Supervised FS utilizes data labels to measure importance and relevance of features. Unsupervised FS, on the other hand, assesses feature relevance by exploiting natural structure of the data without using the class label. Semi-supervised FS benefits from both labeled and unlabeled data. Figure 2 illustrates a taxonomy of grouping-based FS approaches covered in this study. A typical scenario in FS approaches based on grouping is that the features are first partitioned into clusters and then (a) representative feature(s) is (are) selected from each cluster according to a specific metric or technique as shown in Fig. 3.

Figure 2 The representation of feature selection approaches based on grouping.

Figure 3 Typical approach for representative feature selection based on grouping.

Grouping-based feature selection under supervised setting

In the literature, there are many studies that conducted FS through feature grouping. The grouping of features is performed by various techniques including K-means (Chormunge & Jena, 2018), hierarchical clustering (Liu, Wu & Zhang, 2011; Park, 2013), affinity propagation (Harris & Van Niekerk, 2018), graph theories (Yang et al., 2012), information theory metrics (Martínez Sotoca & Pla, 2010), kernel density estimation (Yu, Ding & Loscalzo, 2008), logistic regression (Shah, Qian & Mahdi, 2016) and regularization methods (Petry, Flexeder & Tutz, 2011). With the availability of class labels in datasets, this prevalence is increasing day by day, offering new approaches and gaining new insights into the field.

Several studies performed K-means or hierarchical clustering for grouping features and then they chose genes from each cluster. Sahu, Dehuri & Jagadev (2017) proposed an ensemble approach where K-means is applied first for feature grouping and then three different filter-based ranking techniques (t-test, signal-to-noise ratio (SNR) and significance analysis of microarrays (SAM)) are implemented for each cluster independently; and the feature in the front rank from each cluster is selected to form three distinct feature subsets. Afterwards, features in subsets are subject to additional elimination by checking the inclusion of each feature in other subsets. In other words, a feature is discarded if it is not available in other subsets. They obtain good accuracy for different combinations in general but this study ignores correlations between genes. Another study (Shang & Li, 2016) applied information compression index to group features by hierarchical clustering and then sorted features within each cluster by Fisher criterion measuring the classification capacity of each feature in a cluster. Subsequently, the feature in the front rank is selected for each cluster to form the feature subset.

Regarding selection of features from groups, in addition to ranking, selection can also be performed sequentially. For instance, Zhu & Yang (2013) group features into clusters by a modified affinity propagation algorithm, and then they apply sequential FS for each cluster. Later on, they gather selected features in clusters to acquire the reduced subset. Their experimental results show improvement in execution time and the accuracies are comparable with sequential FS. Alimoussa et al. (2021) proposed a sequential FS method based on feature grouping mainly consisting of three steps. They first remove irrelevant features using Pearson correlation. Then, the same correlation metric is employed for grouping of features into clusters by considering intercorrelated features directly or indirectly via other features. Finally, a feature from each cluster is selected sequentially and features belonging to the same cluster are removed in each round. Their proposed method gives better accuracy and reduction in size compared to filter and wrapper methods. However, despite their approach being fully filter-based, execution time of the proposed method is moderate due to the grouping procedure. In their other work for color texture classification (Alimoussa et al., 2022), they incorporate a classifier into their previous work in order to measure accuracy when a feature is added at each step of their procedure, thereby determining the dimensionality of the feature subset. They show that combining several descriptor configurations performs better compared to a predefined configuration.

Au et al. (2005) proposed an effective algorithm called k-modes attribute clustering algorithm (ACA) for gene expression data analysis. This algorithm uses an information measure to quantify correlation between features, and performs K-mode algorithm, similar to K-means, to cluster features. They defined mode of each cluster as the attribute (i.e., feature) with the highest sum of relevancy with others in each feature group. These modes constituted the final reduced subset. Their measure was also utilized to get good clustering configurations automatically. Chitsaz, Taheri & Katebi (2008) presented a fuzzy variant of this study which relies on the basic underlying idea in fuzzy clustering approaches, that each feature may belong to more than one group. Rather than considering association of each feature with a sole cluster, association with all features among the overall clusters is considered by assigning different grades of membership to features. Their extended work (Chitsaz et al., 2009) integrates chi-square test to assess the dependency of each feature on the class labels during the FS process. In their method, objective function is computed by the following formula (12) J= ∑r=1k ∑i=1purimRAi:ηr

where k and p designate number of clusters and features, respectively and uri is membership degree of ith feature in rth cluster and m is a weighting exponent with ηr being the mode of rth cluster which is essentially center of that cluster. R function denotes interdependence measure between feature Ai and mode ηr. Their experimental results achieve improvement in the accuracy of the classifier with significant reduction in selected feature size compared to the basic version.

Graph-based approaches are also common in studies involving FS through grouping. Song, Ni & Wang (2013) proposed an algorithm, called Fast clustering-bAsed feature Selection algoriThm (FAST), and benefited from minimum spanning trees (MST) to create feature clusters. They adopted SU to determine relevance between any pair of features or between the feature and the target class. Finally, the feature with the highest correlation with the class label is selected from each cluster. Another study (Liu et al., 2014) under supervised framework similarly used MST for grouping and variation of information for relevance measure. Desired number of features and the pruning rate should be given as inputs in their algorithm. A recent study by Zheng et al. (2021) builds the graph by interaction gain, makes use of MST to produce feature groups and probabilistic consistency measure for quality metric including two different techniques for FS: in the first one, they apply the conventional way of selecting representatives from each feature groups; and in the second they use harmony search as a metaheuristic search. The metaheuristic approach dominates their first proposed algorithm together with other search mechanisms. Quite recently, the study proposed by Wan et al. (2023) employs graph theory for feature grouping and selection in a fuzzy space. They initially construct the fuzzy space using neighborhood adaptive β-precision fuzzy rough set (NA- β-PFRS) and then constitute feature groups using MST and acquire the final subset considering feature-to feature and feature-to class relevance in the space. They achieve slightly better results in accuracy with reduced number of features in comparison with other FS approaches and they also show robustness of their model.

Speaking of metaheuristic, García-Torres et al. (2016) employed Markov blanket for clustering features and then these predominant groups are involved in variable neighborhood search (VNS) metaheuristic. Their algorithm yields competitive results in classifier performance and exhibits effective results in terms of number of features and running time. Another optimization-based approach in García-Torres et al. (2021) adopted a scatter search (SS) strategy based on feature grouping where Greedy Predominant Groups Generator (GreedyPGG) (García-Torres et al., 2016) is used to group features. In their metaheuristic approach, each solution generated by the search is enhanced with sequential forward selection for selection of the reduced set of features. Their experimental work shows comparable classification results with SS but a significant reduction in feature subset size. Song et al. (2022a) presents a three-step hybrid study for high dimensional data. Their work initially removes irrelevant features with SU by a predetermined threshold ρ0 which is defined as (13) ρ0= min0.1∗SUmax,SU⌊D/ logD⌋−th

where SUmax is the maximal relevance value between a feature and class labels among all D features. Secondly, it constitutes feature groups using a SU-based clustering approach in which cluster centers are chosen at first and initial number of clusters is not required. As the third step, representative features are selected from clusters based on particle swarm optimization (PSO) with global search capability. Their proposed methodology yields comparative results with respect to accuracy and running time. García-Torres, Ruiz & Divina (2023) extended their previous SS work, integrating an additional stopping criterion into their algorithm along with hyperparameter tuning. Their experimental results present the effectiveness of the additional stopping condition with respect to the computing time, and also exhibit similar classifier performance with highly reduced size of feature subset among other evolutionary and popular approaches.

Although many studies focused their attention on discriminative power and redundancy removal of features, most of them neglect the stability of the selected features. Yu, Ding & Loscalzo (2008) addressed this issue in their two studies. In Yu, Ding & Loscalzo (2008), rather than relying on typical clustering algorithms, they applied kernel density estimation accompanied by an iterative mean shift procedure to find feature clusters. Subsequently, these feature clusters were evaluated according to relevance using F-statistic and a representative feature is selected within each cluster. The same authors extended this study in Loscalzo, Yu & Ding (2009), where consensus feature groups were identified in an ensemble learning manner and features were extracted in the same way as their first study. The experiments conducted in both studies showed the stability of the selected features.

All the works mentioned until now are considered as global FS, i.e., finding a reduced subset of global features for the entire population. However, there are cases where these approaches are not applicable. For instance, take an image recognition task, where feature importance may alter since a set of relevant features may be important for identifying a specific object but insignificant for another object at a different position. This gap paved the way for a different technique, called Instance-wise FS that associates each feature’s relationship to its labels by assigning a different selector for each instance. Interested readers to grouping and selection of features in this approach can refer to (Xiao et al., 2022; Masoomi et al., 2020). A summary of above-mentioned approaches under the supervised framework is outlined in Table 1.

Table 1 Applications of FS by grouping under supervised context.

	Grouping Method	FS Method (metric)	FS Strategy	Validation	Types of Data	Study	
K-means	correlation	selection of features from front rank	classification accuracy	text and microarray	Chormunge & Jena (2018)	
		SNR, SAM, t-test	checking existence of a feature in other subsets	leave one out cross validation (LOOCV)	microarray	Sahu, Dehuri & Jagadev (2017)	
Hierarchical	Fisher	selection of features from front rank	classification accuracy	miscellaneous	Shang & Li (2016)	
		average similarity	choosing representative in each group	cross validation	miscellaneous	Park (2013)	
Sequential	Correlation-based	trace criterion	features are added sequentially only when trace is maximum.	cross validation	color texture	Alimoussa et al. (2022)	
	Modified Affinity Propagation	sequential feature selection	applying sequential search in each group and merging selected features	cross validation	miscellaneous	Zhu & Yang (2013)	
ACA	interdependence mesure	selection of mode of each cluster	classification accuracy	synthetic & gene expression	Au et al. (2005)	
Fuzzy	Correlation	fuzzy-rough subset evaluation	selection of representative features among groups in the fuzzy environment	classification accuracy	miscellaneous	Jensen, Parthalain & Cornells (2014)	
	Fuzzy ACA	fuzzy multiple interdependence redundancy		classification accuracy	miscellaneous	Chitsaz et al. (2009)	
		fuzzy multiple interdependence redundancy		classification accuracy	microarray	Chitsaz, Taheri & Katebi (2008)	
Graph-based	neighborhood adaptive fuzzy mutual information	using feature-to-feature & feature-to-class relevance	cross validation	publicly available datasets	Wan et al. (2023)	
		probabilistic consistency	(i) choosing representative in each group (ii) metaheuristic search	cross validation	miscellaneous	Zheng et al. (2021)	
		variation of information	choosing representative in each group	silhoutte index & classification accuracy	miscellaneous	Liu et al. (2014)	
		SU	choosing representative in each group	classification accuracy	miscellaneous	Song, Ni & Wang (2013)	
Evolutionary	GreedyPGG	SS	using SS to find subset of features	cross validation	gene expression & text-mining	García-Torres, Ruiz & Divina (2023)	
	SU-based	PSO	adopting PSO to determine final subset	cross validation	miscellaneous	Song et al. (2022a)	
	GreedyPGG	SS	using SS to find subset of features	cross validation	biomedical datasets	García-Torres et al. (2021)	
	GreedyPGG	VNS	utilizing VNS to decide reduced subset	cross validation	microarray & text-mining	García-Torres et al. (2016)	

FS approaches based on grouping are not necessarily in the manner of grouping features into clusters and choosing representatives. Distinctly, selection of the features may happen with different cluster configurations. Moslehi & Haeri (2021) initially implement K-means for clustering all samples for a given dataset and a sample from each cluster is chosen at random to acquire the samples with the greatest differences for the preliminary dataset. Subsequently, variances of all features on the determined samples are calculated and a predefined number of features with the highest variances are selected, thereby forming the primary dataset. Thereafter, remaining features are added gradually to this dataset and K- means clustering with a predefined number of clusters is applied iteratively in each step. Features causing changes in the structure of clusters are observed in a repetitive manner and considered as significant. Other features that don’t lead to any alteration in clusters are eliminated.

Another work by Yousef et al. (2007) introduced the “recursive cluster elimination” term into the community and their approach was later adopted in many studies. Since this approach was widely employed by different studies, in ‘Feature Grouping with Recursive Cluster Elimination’ we elaborate this method in detail by reviewing its application areas and modified usages.

Grouping-based feature selection under unsupervised setting

As with the traditional methods in FS, many of feature grouping-based FS approaches belong to the supervised learning paradigm. Unsupervised FS is more challenging than supervised FS because of no prior knowledge about class labels and unknown number of clusters. Unsupervised FS methods typically involve (i) maximization of clustering performance by some index or (ii) selection of features based on dependency. Since this article is about FS, first one is out of scope for this study. Many statistical dependency/distance measures are available in the literature including correlation coefficient, least square regression error, Euclidean distance, entropy, and variance. Selected features in unsupervised FS methods can be evaluated in terms of both classification performance and clustering performance. Table 2 summarizes works on unsupervised FS based on grouping.

Table 2 Applications of FS by grouping under unsupervised context.

Grouping Method	FS Method (metric)	FS Strategy	Validation	Types of Data	Study	
K-means	generalized incoherent regression model	grouping and selection of optimal features based on orthogonal constraints	unsupervised clustering accuracy (ACC) & normalized mutual information (NMI)	face image & biological datasets	Yuan et al. (2022)	
Louvain community detection	BAS	features in each group are sorted by modified BAS and best features are selected iteratively	classification error rate (CER)	real-world datasets	Manbari, AkhlaghianTab & Salavati (2019)	
SU-based	SU	feature with the highest SU on average is chosen as representative in each cluster	scatter separability criterion, random adjust index, normalized mutual information, F-score	miscellaneous	Zhu et al. (2019)	
K-mode	mode	selection of mode of each cluster	classification accuracy	miscellaneous	Zhou & Chan (2015)	
Affinity Propagation	MICAP	centroid of each cluster is selected for final subset	classification accuracy	miscellaneous	Zhao, Deng & Shi (2013)	
k-medoids	Simplified Silhouette Filter (SSF)	medoid of each cluster is chosen as the representative feature	classification accuracy	miscellaneous	Covões et al. (2009)	
hierarchical	FS through Feature Clustering (FSFC)	feature with the shortest distance to others is selected in each cluster	Minkowski Score	public gene datasets	Li et al. (2008)	
kNN	entropy	a single feature from each cluster is chosen applying entropy	entropy, fuzzy feature evaluation index, classification accuracy	real life public domain	Mitra, Murthy & Pal (2002)	

Mitra, Murthy & Pal (2002) proposed an unsupervised FS algorithm using feature similarity. A new similarity measure called maximum information compression index is introduced in their study. Also, they demonstrated use of representation entropy for measuring redundancy and information loss quantitatively. Features are partitioned into clusters using K-nearest neighbors (KNN) principle along with a similarity measure. Entropy metric is chosen as the FS criterion and applied to select a single feature from each cluster to constitute the reduced subset. To evaluate the effectiveness of selected features, the proposed method is compared with KNN, naive Bayes and class separability including Relief-F for classification capability, and with entropy and fuzzy feature evaluation index for clustering performance. Their algorithm is rapid since no search is required and hence their study is one of the state of the art work in the literature.

Another example is the study of Li et al. (2008), which uses the same similarity measure in Mitra, Murthy & Pal (2002) and employs a distance function to obtain clusters of features. A representative feature, having the shortest distance to others within a cluster, is selected from each cluster. Their approach is based on hierarchical clustering which enables them to choose feature subsets with different sizes by choosing from top clusters in the hierarchy. Their algorithm works for both unsupervised and supervised learning tasks. Moreover, they run clustering just one time in their algorithm. The authors presented their experimental results for both clustering and classification.

As stated previously, FS methods developed under unsupervised framework do not utilize class labels. As an example, Covões et al. (2009) presents a comparative study of their approach with the algorithm proposed by Mitra, Murthy & Pal (2002). Again, maximal information compression index is utilized to find clusters of features. Hereafter, they employed the simplified silhouette criterion to find optimum clusters, allowing to find the number of clusters as well. The computation for simplified silhouette depends only on obtained partitions, and it is not dependent on any clustering algorithm. Hence, this silhouette is, not only determines the number of clusters automatically, but also it is capable of evaluating partitions acquired by any clustering algorithms. They employed the k-medoids algorithm along with the silhouette method in order to achieve optimum clusters. Then the corresponding medoid for each cluster is selected as the representative feature. The prerequisite for number of clusters known a priori in this algorithm has been overcome by the simplified silhouette since one can implement this algorithm for different values of number of clusters, and then select the best clustering according to the maximum value obtained in the silhouette.

Another study under unsupervised framework is suggested in Zhao, Deng & Shi (2013), where maximal information coefficient and affinity propagation (MICAP) are exploited for selection of features. Features are chosen as the centroid of each cluster in the final step. Although they present competitive results in classification with typical classifiers, no comparison is made for clustering.

FS methods developed under supervised framework can be an inspiration to unsupervised studies. For instance, Zhou & Chan (2015) developed an attribute clustering algorithm along with an FS method in an unsupervised manner. They test their algorithm considering different FS methods with different classifiers and achieve slightly improved mean accuracies. The unsupervised FS algorithm proposed by Zhu et al. (2019) groups features according to their SU similarities. In their clustering approach, cluster centers are firstly determined and the features are assigned to these centers subsequently. Then, the feature with the highest SU on average is selected from each cluster as a representative based on the following formula (14) ARf,C=∑i=1|C|SUf,fi|C|

where ARf,C is the average redundancy for a feature f in cluster C and fi ∈ C. Their experiments showed that compared to other methods, the proposed algorithm performs more efficiently in terms of running time and in terms of the size of the reduced subset of features. Also, clustering performance of their algorithm surpasses the compared techniques for various clustering performance measurements. Apart from this, a recent hybrid work which is a combination of grouping and binary ant system (BAS) can be found in Manbari, AkhlaghianTab & Salavati (2019).

More recently, Yuan et al. (2022) formulated this phenomenon as an optimization problem, where their optimization benefits from feature grouping and orthogonal constraints. Clustering performance of their algorithm shows better performance in general compared to other unsupervised FS methods.

Grouping-based feature selection under semi-supervised setting

There are cases when a significant amount of data is unlabeled and only few samples are labeled. In such a case, the learning problem is denominated as semi-supervised. Quinzán, Sotoca & Pla (2009) conducted a grouping-based FS study under this setting. In their study, the distance measure between each pair of features is computed by both conditional entropy and conditional mutual information. Next, hierarchical clustering is applied to attain feature clusters and the feature with the highest MI is selected as the representative inside each cluster. They test the performance of their algorithm for a different number of labeled samples with other algorithms and their results exhibit satisfactory performance when there is not enough labeled data. Semi-supervised FS techniques are common in the literature and reviewed in many studies (Song et al., 2022b; Kostopoulos et al., 2018; Sheikhpour et al., 2017).

Feature Grouping with Recursive Cluster Elimination

In the original framework (Yousef et al., 2007), the first step in SVM-RCE is to group genes (i.e., features) into clusters using K-means in which correlated gene clusters are identified. As the second step, SVM is used to score and rank these clusters and finally clusters with low scores are eliminated. Remaining genes in clusters are combined and then clustering along with SVM is applied iteratively until a predefined number of clusters are left. In each iteration, surviving genes are used for classification to measure the accuracy at each level. Interests in this method have grown rapidly over time and many studies conducted their research via integrating this approach. The schematic diagram of this approach is illustrated in Fig. 4.

Figure 4 The workflow of the SVM-RCE algorithm.

The grouping step for grouping genes into clusters, the scoring step for assigning score for each cluster and selecting significant clusters, the modeling step for training the model with top-ranked clusters.

Weis, Visco & Faulon (2008) presented a SVM-RCE-like approach where they included assessment of clusters collaboratively rather than evaluating clusters individually. The study of Deshpande et al. (2010) utilized SVM-RCE with small modifications for brain state classification.

Another study by Luo et al. (2011) aimed to reduce the computational complexity of SVM-RCE. They apply infinite norm of weight coefficient vector from the SVM model to score each cluster instead of scoring clusters by cross-validation. Their results show considerable reduction in computation time while exhibiting comparative performance as SVM-RCE.

In the study associated with military service members, in addition to the statistical significance test, SVM-RCE is used to classify individuals between posttraumatic stress disorder (PTSD), postconcussion syndrome (PCS) + PTSD, and controls (Rangaprakash et al , 2017). In their study, the features refer to the connectivity paths acquired from 125 brain regions. In their experimental works using SVM-RCE, they conclude that higher classification rate (by 4%) is achieved through imaging-based grouping than conventional grouping. Furthermore, imaging measures dominate non-imaging measures by 9% for both conventional and imaging-based groupings.

Jin et al. (2017) conducted a similar study and adopted a modified version of SVM-RCE in their study of brain connectivity. In their study, the diagnostic label of a novel subject is tested whether it belongs to subjects with PTSD or healthy group. The connectivity features are measured from mean resting-state time series taken from 190 regions across the entire brain. They employ SVM-RCE in their experimental work to suggest that dynamic functional and effective connectivity gives higher classification results compared to their static counterparts.

Interestingly, Zhao, Wang & Chen (2017) applied SVM-RCE tool to the detection of expression profiles identifying microRNAs related to venous metastasis in hepatocellular carcinoma.

Chaitra, Vijaya & Deshpande (2020) conducted a study to identify biomarkers of autism spectrum disorder (ASD) using imaging datasets. They utilized SVM-RCE to assess the classification performance for three distinct feature sets consisting of connectivity features alone, complex network (i.e., graph) measures alone, and a feature set including both. Their accuracy results are not competitive; however, the emphasis is on assessing different feature sets, especially on the combined feature set.

Grouping Features with Biological Domain Knowledge

The aforementioned FS approaches typically apply statistical analysis and run computational algorithms to create the feature groups. Hence, these approaches are fully data-driven and they generate the groups of features without using any domain knowledge. However, in some fields, the automatic transformation of data into information via exploiting the background knowledge in the domain is very beneficial. Background knowledge refers to the domain knowledge obtained from the literature, domain experts or from available knowledge repositories (Bellazzi & Zupan, 2007). In such fields, the integration of domain knowledge into the feature selection process might improve performance, and also might reveal novel knowledge. For example, in the field of bioinformatics and computational biology, the integration of biological domain knowledge is used to improve the process of feature selection (i.e. gene selection in gene expression data analysis, in other words biomarker discovery) (Perscheid, 2021; Yousef, Kumar & Bakir-Gungor, 2020).

This section deals with how feature groups are created and how FS is realized using biological external sources. The main idea behind the integration of biological knowledge to FS is to apply a biological function to create groups of features (i.e., groups of genes) and then employ a learning algorithm to score these generated groups. Finally, the genes in the top scoring groups form the reduced subset of features. We would like to note that this section is especially designed for researchers working in the field of molecular biology, genetics, bioinformatics; and we believe that this section is especially informative for those with a biological background. Still, scientists working in different fields can get inspiration from the studies presented in this section and apply similar domain knowledge-based feature grouping in their problems. For example, in the field of text mining, a related tool named TextNetTopics (Yousef & Voskergian, 2022) uses Latent Dirichlet Allocation (LDA) to detect topics of words, which serve as groups of features.

As one of the pioneers in this field, Bellazzi & Zupan (2007) discussed the shift of gene expression data analysis approaches from purely data-centric approaches to integrative approaches which aim at complementing statistical analysis with knowledge acquired from diverse available resources. The authors reported that with the growing number of knowledge bases, the field has shifted from purely data-oriented methods to methods that aim to include additional knowledge in the data analysis process (Bellazzi & Zupan, 2007). The authors presented the modifications of clustering algorithms for embedding background knowledge. More specifically, the authors provide a survey of approaches that adapt distance-based, model-based and template-based clustering methods so that they take the additional background knowledge into account.

Yet as another review article in this field, recently Perscheid (2021) published a survey on prior knowledge-based approaches for biomarker detection through the analysis of gene expression datasets. In that article, she evaluated the main characteristics of different integrative gene selection approaches; and she presented an overview of external knowledge bases that are utilized in these approaches (Perscheid, 2021). It is reported that Gene Ontology (GO) (Ashburner et al., 2000) and Kyoto Encyclopedia of Genes and Genomes (KEGG) (Kanehisa, 2000) resources are predominantly used as external knowledge bases for integrative gene selection. The author classified existing integrative gene selection approaches into three distinct categories (i.e., modifying approaches, combining approaches, module extraction approaches). The same review article presented a qualitative comparison of existing approaches and discussed the current challenges for applying integrative gene selection in practice via pointing out directions for future research. An interested reader can refer to Perscheid (2021) for further details.

As one of the biological knowledge-based feature grouping approaches, Support Vector Machines with Recursive Network Elimination (SVM-RNE) (Yousef et al., 2009), was proposed as an extension of SVM-RCE, which is presented in the previous section. In Yousef et al. (2009), genes are grouped into clusters using Gene eXpression Network Analysis (GXNA) (Wang et al., 2007) and clusters with low scores are eliminated in each iteration. The algorithm terminates when some predefined constraints on the number of groups are met.

As another biological knowledge-based integrative approach, Qi & Tang (2007) attempt to incorporate GO annotations into the gene selection process, where they start by finding a discriminative score for each gene (i.e., feature) via applying IG, and eliminating those with a score of zero. The next step is to annotate these genes with GO terms. After that, the score of each term is calculated as the mean of discriminative scores of associated genes involved in the respective term. The GO term with the highest score is determined and the most discriminative associated gene is selected and extracted. The steps including calculation of scores for GO terms and selection of the next most informative gene is repeated until the final subset is formed. Their comparative results with only using IG shows the effectiveness of GO integration in the gene selection process (Qi & Tang, 2007). Some other approaches for biological data integration include Bayesian methods, tree-based and network-based techniques (Li, Wu & Ngom, 2016).

Incorporating biological knowledge in the clustering algorithm is reported as a very challenging task (Perscheid, 2021). Along this line, the GOstats package (Falcon & Gentleman, 2007) allows one to define semantic similarity between the genes via incorporating the GO. As another example of domain knowledge-based gene selection, in SoFoCles (Papachristoudis, Diplaris & Mitkas, 2010), genes are initially ranked by typical filter methods such as IG, Relief-F or χ2, and then a reduced subset of genes is created using a predefined threshold. Next, for each gene in the reduced subset, semantically similar genes from GO are determined. Finally, top semantically similar genes are selected to enrich the reduced subset. Experimental works conducted using SoFoCles reveal enhancement in classification results by integrating biological knowledge into gene selection.

An additional study by Mitra & Ghosh (2012) adopted the Clustering Large Applications based on RAN-domized Search (CLARANS) technique to gene (i.e., feature) clustering via utilizing GO analysis. In Mitra & Ghosh (2012), the final reduced feature subset is composed of the genes which were medoids of biologically enriched clusters. Their experimental results showed that the incorporation of biological knowledge enhanced classifier performance and reduced computational complexity. The same authors subsequently made use of a fuzzy technique, Fuzzy Clustering Large Applications based on RAN-domized Search (FCLARANS), to obtain clusters and they selected representative genes from clusters based on the fold change (Ghosh & Mitra, 2012).

The study suggested by Fang, Mustapha & MdSulaiman (2014) utilizes both KEGG and GO terms with IG. In Fang, Mustapha & MdSulaiman (2014), IG is applied on the initial dataset as filtering and then GO and KEGG annotations are explored for the remaining genes. As the next step, association mining is applied to this annotation information and the interestingness of the frequent itemsets is determined by averaging the original discriminative scores of the involved genes. The final gene set is attained via the selection of the highest ranked genes from the top n frequent itemsets. They assessed their method using GO, KEGG, and both against IG and study of Qi & Tang (2007). Despite the lower rate of improvement in the overall accuracy, they are able to achieve the increase in accuracy with a significant reduction in the number of genes.

Yet as another domain knowledge-based gene selection approach, Raghu et al. (2017) utilize the KEGG (Kanehisa, 2000), DisGeNET (Piñero et al., 2019) and other genetic meta information in their integrated approach. In their framework, two metrics, i.e., gene importance and gene distance, are computed. Importance score for each gene is calculated using DisGeNET, which is a public platform containing gene collections associated with diseases. Distance between genes is computed based on their chromosomal locations and associations to the same diseases. Both scores are then employed to compose gene sets with maximum relevance and diversity. Compared to variance-based techniques, their method performs better in the predictive modeling task on a small scale.

Another related study developed maTE tool (Yousef, Abdallah & Allmer, 2019), where gene groups are created based on the miRNA-target gene information, and then each group is ordered by cross-validation. The average accuracy after a specific number of iterations determines the rank of each cluster. Genes on the top m groups are selected as the reduced subset (Yousef, Abdallah & Allmer, 2019).

As another example, the Grouping-Scoring-Modeling (G-S-M) method benefits from the biological knowledge for its grouping step, followed by the ranking and classification steps (Yousef, Kumar & Bakir-Gungor, 2020). Following the G-S-M approach, CogNet framework (Yousef, Ülgen & Uğur Sezerman, 2021) initially implements pathfindR (Ulgen, Ozisik & Sezerman, 2019) to group the genes. The genes in each group are actually the genes of an enriched KEGG pathway, identified as a result of the active subnetwork search and functional enrichment steps of pathFindR. Then, a new dataset involving genes for the specific pathway is created for each group (i.e., pathway). These datasets are scored through Monte Carlo cross-validation (MCCV) and the pathways are ranked according to the assigned scores. Ultimately, genes found in top chosen pathways are taken as selected features and they are used for classification. Another study, developed the miRcorrNet tool (Yousef et al., 2021b), which finds gene groups on the basis of their correlation to miRNA expression. Afterwards, these groups are subject to a ranking function for classification. The results showed area under curve (AUC) scores above 95%, proving that miRcorrNet is capable of prioritizing pan-cancer-regulating high-confidence miRNAs. The G-S-M approach has been used by other bioinformatics tools. An example of such tools are: miRModuleNet (Yousef, Goy & Bakir-Gungor, 2022), which detects groups via calculating the correlations between the mRNA and miRNA expression profiles; Integrating of Gene Ontology (Yousef, Sayıcı& Bakir-Gungor, 2021) that uses Gene Ontology information for grouping; PriPath (Yousef et al., 2023) that uses KEGG pathways for grouping; GediNet (Qumsiyeh, Showe & Yousef, 2022) that uses disease gene associations as groups; 3Mint (Unlu Yazici et al., 2023) that employs mRNA expression, miRNA expression and methylation profiles for grouping; and miRdisNET (Jabeer et al., 2023) that uses miRNA target gene information while creating the groups.

Very recently Zhang et al. (2022) proposed a method called Distance Correlation Gain-Network (DCG-Net); where they quantify distance correlation gain between features to construct the biological network. In their algorithm, a greedy search method is applied to detect network modules. The edge with the highest weight is selected, then this edge is extended with respect to correlation metric to obtain the module in the network. This is done iteratively to extract modules and the module with the highest distance correlation is selected for analysis. Their experimental results showed effective results in terms of FS and classification accuracy.

Perscheid, Grasnick & Uflacker (2018) comparatively evaluated traditional gene selection methods with knowledge-based methods. Their approach produces gene rankings by integrating knowledge bases and each of these rankings are evaluated with a predefined number of selected genes. Finally, the ranking with the best performance is selected. Moreover, they proposed a framework allowing external knowledge utilization, gene selection and evaluation in an automatic fashion. Although the framework seems to be knowledge base dependent, their experimental results demonstrate that incorporating biological knowledge into the gene selection process improves classification performance, decreases computational running time, and enhances the stability of selected genes.

Discussion

As stated previously, FS based on feature grouping is a powerful technique with important advantages. Next, one may wonder which FS technique is the best in this context. Surely, it’s hard to answer this question because the concept of FS is not dependent only on one parameter. The intrinsic structure and size of the dataset, the learning model and the selected parameters are known as effective factors in the field. In this section we make a cross-comparison and share our deductions among the approaches we have examined in the literature.

We mentioned before that a typical approach in grouping-based FS is to select representative features from groups. However, selection of multiple representatives from groups may enhance the classifier performance as shown in Covões & Hruschka (2011). In Covões & Hruschka (2011), the least correlated feature with other features in the same cluster is selected in addition to the selection of the representative. Hence, higher accuracy values are achieved.

The superiority of feature grouping is apparent in sequential-based FS because once a feature is selected, features of the same cluster can be discarded at each iteration, thereby diminishing search complexity in total. We particularly want to emphasize here that sequential-based FS approaches generally employ wrapper models which cause huge running time. We motivate researchers for filter-based sequential FS techniques since such an approach benefits both from the strength of feature grouping and from the high speed of filter models as presented in Alimoussa et al. (2021); Alimoussa et al. (2022). Dominance of this approach over deep learning algorithms can be seen in Alimoussa et al. (2022). As a result, sequential approaches are effective in the field since they consider interactivity between features and are also used during subset search in evolutionary approaches (García-Torres et al., 2016; García-Torres, Ruiz & Divina, 2023).

Fuzzy approaches for FS based on grouping are effective because features can belong to more than one cluster rather than typical assignment of a feature to a specific cluster, which can improve the subset quality and accuracy. We should also say that feature-class relevance is an important metric in supervised setting for fuzzy or other approaches and importance of its utilization is specified in Chitsaz et al. (2009). On the other hand, evolutionary algorithms such as genetic algorithms can be implemented as subset search algorithms during the selection process (Lin et al., 2015). These approaches outperform the conventional way of selecting representatives due to inclusion of inter-feature collaboration as shown in Zheng et al. (2021). The main challenge for these algorithms is their high computational cost. A comparison of fuzzy and evolutionary approaches is available in Jensen, Parthalain & Cornells (2014), where both methods obtain similar accuracies but the proposed fuzzy technique dominates others in terms of running time and subset quality.

Incorporating different techniques can increase the strength of an approach rather than sticking to a specific one alone. For instance, the study of Wan et al. (2023) combines the advantages of fuzziness, graph theory and conditional mutual information, and acquires better results in general than graph-based or fuzzy approaches.

As implied in ‘Grouping Features with Biological Domain Knowledge’, integrative gene selection is an important matter when biological data is considered since statistical methods lack the ability to identify the underlying biological processes. Effectiveness of integrating domain knowledge from external sources is reviewed in Perscheid (2021) and Perscheid, Grasnick & Uflacker (2018).

FS methods based on deep learning (DL) are common in the literature (Hassan et al., 2022; Hussain et al., 2022; Krell et al., 2022) but these methods adopt feature extraction, i.e., transformation of the original feature space into a reduced size of new features which leads to loss of original semantics of features. In short, they provide competitive class accuracies but are far from interpretability (Figueroa Barraza, López Droguett & Martins, 2021).

Despite the plenitude of FS techniques, there is still room for further progress in this field. The current studies are mostly based on pairwise interactions; whereas interactions of multiple features should be explored. In addition, running time is still a barrier, and especially for complex algorithms smart steps should be taken on it.

Conclusions

The advances in high-throughput technologies have generated large high-dimensional data sets in many applications. The inevitable presence of redundant and noisy features increases computational complexity and degrades classifier capability. Hence, FS has become a required pre-processing step in itself as a primary concern for a long time. Here, we present works done in the literature regarding FS techniques through feature grouping. Feature grouping is a powerful and efficient concept; it reduces search space and complexity, is resistant to the variations of samples, gives lower levels internal redundancy and provides better generalization capability to the classifier. The form of feature grouping and selection of features out of groups are determined by different metrics or techniques as reviewed in this article.

In FS-based feature grouping, the aim is to first keep similar features together within clusters while maximizing diversity between clusters followed by selection of features out of clusters. We can conclude that sequential and optimization-based (i.e., fuzzy and evolutionary) FS approaches are noteworthy in this context since they take feature interactivity into consideration during the selection phase. Hybrid approaches or utilizing a combination of different techniques are also effective because each method brings its advantage. In the case of biological data, integrating external knowledge can yield better results in the overall analysis. In fact, the availability of independent and relevant features, correlation between features, and feature correlation to the decision are important items to be taken into consideration. The models with the ability to take these factors into consideration are likely to be effective in FS.

In this study, our goal is to inform interested readers about the recent trends in FS by feature grouping. Despite the wealth of many techniques in this field, there is still need for enhancement and novelty in the area. We believe approaches mentioned here may provide new insights into designing new schemes for FS in terms of better efficiency, effectiveness, stability, generalization and discrimination.

Additional Information and Declarations

Competing Interests

Author Contributions

Data Availability

Burcu Bakir-Gungor is an Academic Editor for PeerJ.

Cihan Kuzudisli conceived and designed the experiments, performed the experiments, analyzed the data, prepared figures and/or tables, authored or reviewed drafts of the article, and approved the final draft.

Burcu Bakir-Gungor conceived and designed the experiments, performed the experiments, analyzed the data, authored or reviewed drafts of the article, and approved the final draft.

Nurten Bulut performed the experiments, analyzed the data, prepared figures and/or tables, and approved the final draft.

Bahjat Qaqish performed the experiments, analyzed the data, authored or reviewed drafts of the article, and approved the final draft.

Malik Yousef conceived and designed the experiments, prepared figures and/or tables, authored or reviewed drafts of the article, and approved the final draft.

The following information was supplied regarding data availability:

This is a literature review.

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
