# Peer review of "Review of feature selection approaches based on grouping of features"

_PeerJ, doi:10.7717/peerj.15666_

## Round 0.1 · original submission · Major Revisions

Feature selection and grouping is indeed an important step in integrating high dimensional datasets. This is an interesting article and I have few suggestions:

1. Line 100-102; Revise this sentence so that its more clear to the audience.
2. Under the section “Rationale of the review and intended audience” or may be in a separate section: The authors should also include a paragraph describing different types of biological data where feature selection process can be applied. For examples, microarray, genomics, epigenomics, transcriptomics etc. This will be helpful for the readers with a biological background to get an idea of how they can implement this process into their data integration analysis.
3. Line 120: “have emerged late 90s” should be “have emerged in late 90s”.
4. Line 174: Revise the sentence: “This method can also be used to measure correlation on a feature-feature basis in order to remove redundant features.”
5. Line 138: It should be Figure 2 (not Figure 1)
6. No references are cited in section 2.2: wrapper method and section 2.3: Embedded method. Please cite the appropriate references.
7. Line 335: cite the appropriate reference here.
8. Figure 2: It’s hard to see which is A, B and C. Label them properly.

Please note: All the reviewers comments/suggestions need to be adequately addressed before further consideration can be made.

Reviewer 1 has suggested that you cite specific references. You are welcome to add any of them if you believe they are relevant. However, you are not required to include these citations, and if you do not include them, this will not influence my decision.

Reviewer 1 ·

Basic reporting

The work is very interesting and I think it requires some improvements. No 2022 references were found. It is necessary to include more recent works. Some of these works are:

- Yuan, A., Huang, J., Wei, C., Zhang, W., Zhang, N., & You, M. (2022, August). Unsupervised Feature Selection via Feature-Grouping and Orthogonal Constraint. In 2022 26th International Conference on Pattern Recognition (ICPR) (pp. 720-726). IEEE Computer Society.
- Wan, J., Chen, H., Li, T., Sang, B., & Yuan, Z. (2022). Feature grouping and selection with graph theory in robust fuzzy rough approximation space. IEEE Transactions on Fuzzy Systems.
- García-Torres, M., Ruiz, R., & Divina, F. (2023). Evolutionary feature selection on high dimensional data using a search space reduction approach. Engineering Applications of Artificial Intelligence, 117, 105556.
- García-Torres, M., Gómez-Vela, F., Divina, F., Pinto-Roa, D. P., Noguera, J. L. V., & Román, J. C. M. (2021, July). Scatter search for high-dimensional feature selection using feature grouping. In Proceedings of the Genetic and Evolutionary Computation Conference Companion (pp. 149-150).
- Hassan, M. M., Mollick, S., & Yasmin, F. (2022). An unsupervised cluster-based feature grouping model for early diabetes detection. Healthcare Analytics, 2, 100112.
- Dai, Y., Gao, Z., Zhu, Y., Zhang, W., Li, H., Wang, Y., & Li, Z. (2022, July). Feature Grouping for No-reference Image Quality Assessment. In 2022 7th International Conference on Automation, Control and Robotics Engineering (CACRE) (pp. 204-208). IEEE.
- Sood, M., Angra, P., Verma, S., & Jhanjhi, N. Z. (2022). Efficient Feature Grouping for IDS Using Clustering Algorithms in Detecting Known/Unknown Attacks. In Information Security Handbook (pp. 103-116). CRC Press.
- Prathiba, T., & Kumari, R. (2022). Retraction Note to: Content based video retrieval system based on multimodal feature grouping by KFCM clustering algorithm to promote human–computer interaction. Journal of Ambient Intelligence and Humanized Computing, 1-1.
- Krell, E., Kamangir, H., Friesand, J., Judge, J., Collins, W., King, S. A., & Tissot, P. (2022). The influence of grouping features on explainable artificial intelligence for a complex fog prediction deep learning model.

Experimental design

- I would suggest to include the symmetrical Uncertainty measure in Section 2.1

- To make the reading easier I would suggest to include a schematic diagram about the different approaches that have been used in feature grouping

Validity of the findings

no comment

·

Basic reporting

This is a study that discusses different feature selection approaches and the authors have gathered details about these feature section approaches. The paper is generally well-written and structured. However, in my opinion, the paper has some shortcomings in regard to the text and contents. The introduction/background is well-written and clear. However, the manuscript is missing references in many places. For instance, under the introduction, in paragraph 3, "Some studies clustered samples (observations) for improving classification performance but were not concerned with feature reduction at all." - there are no references given to show "some studies". Some other examples are:
section 2: there is no reference given for filter methods, information gain, Pearson's correlation, chi-square, etc. So I suggest the authors add references throughout the manuscript.

Consistency is missing in some parts of the manuscript. For instance: under Introduction, paragraph 4, "Hereafter, grouping and clustering terms will be used interchangeably" - using either grouping or clustering would help the readers to follow the manuscript easily.

It would be great to add more details from the cited papers. for instance: under 3.1: "A comparative work on various filtering methods (mixture model, regression modelling and t-test) was proposed in [9] and they outlined similar and dissimilar aspects of these methods. Lazar et al. [10] also reviewed filter-type FS algorithms used in gene expression microarray analysis." - if the authors add more details like the outcome of the study, it would help the readers to get an idea about their work.

Given these shortcomings the manuscript requires major revisions.

Experimental design

This is a well-structured and well-designed study. However, in my opinion, the paper has some shortcomings regarding the contents of the manuscript. There are so many FS methods added to the review. It would be great to have deeper details about each method. Information gain, Peterson's correlation, etc could have more details added to it.

Also, none of these methods are having proper references.

Not all variables are defined from the equations given in the manuscript. For instance: under 2.1.1: equation 1 - "n" is not defined.
2.1.2: equation 2 - there is no variable named C in the equation, also r is not defined.
2.1.3: equation 3 - m is not defined, etc
So I would suggest reviewing these equations and add define all the variables.

Under 2.1.3: the symbol used for chi-square is not consistent.

Validity of the findings

The manuscript is a review of different FS methods. There are no new findings involved in this study. However, the reviewed FS methods need to have more information added to them. The conclusion need to be improved.

Additional comments

The study reviewed multiple FS methods and is a well-structured study. However, the reviewed methods need to have more details. Maybe, the authors may apply these methods to a dataset and present the output. Thereby, the authors can suggest appropriate FS methods to the reader for different data types and situations.

There can be a comparison performed between the FS methods selected for review. Thereby the conclusion part can be improved.

Reviewer 3 ·

Basic reporting

It is a review on feature selection methods based on data grouping. I have the following concerns/suggestions:
Major suggestions:
1) What was the query string used to execute the query in the considered literature repositories such as WoS, Scopus, and Google Scholar? It must be mentioned along with the date of executing query.
2) State including and exclusion criteria of the literature clearly.
3) Feature Selection under Unsupervised and Supervised Settings has been discussed. Authors also need to add feature selection under semi-supervised (hybrid) settings.
4) There is no cross-comparison among the discussed methods. Authors need to compare and contrast them, and mention the scenarios where each one can be a better choice.
5) What about feature-selection free methods of learning, specially deep learning methods where feature engineering is not required? Authors need to add in the "Discussion" section, including limitations of the current feature selections methods, and future directions.
6) Table 1 and Table 2 need more explanation. How k-means may come in Supervised settings (ref. Table 1)?
7) How can fuzzy (ref Table 1) can be directly used for feature selection? Authors need to add other methods that are combined with fuzzy for the purpose, if any.
8) In Table 1 column headings, "Application area" is inappropriate here. It may be "Types of Data". Authors may add one more column to this Table to mention some descriptions. Currently, it is very hard to understand Table 1 and Table 2.
9) Figure 1 is not a generic diagram in the current context. Figure is focused on the grouping of genes only. I would suggest to make it generic in nature, as per the title of the paper.
10) Figure 1 and Figure 2 need more explanation within the text.

Some of the minor suggestions are:
11) Some of the citation are not in the correct way and not in sequence. It should be arranged in proper order. Ex: [26], [27] it should be like [26-27].
12. In entire section 2, Author have not cited even a single paragraph.
13. In section 2.3,3.2.1, 3.2.1, 3.2.2 there is no any mathematical equation. It will be better if include some mathematical equation.
14. Authors have used very less recently published papers.
15. Section 5 content is not informative. Manuscript writing style should be improved in this section.

Experimental design

The methodological design of the review paper needs minor improvement as suggested in Basic Reporting section.

Validity of the findings

No comments

Additional comments

No comments

---

## Round 0.2 · Minor Revisions

Please address the remaining reviewer comments.

In addition, I have minor comments:

1. Provide high quality figures for publication.
2. Line 156; Remove “field” in the sentence “In this section, we present basic concepts in the FS field”.
3. Line 399-407: Cite appropriate references in the paragraph.
4. Provide full form for followings and make sure full forms are provided throughout the manuscript.
a. Line 437; SAM
b. Line 488;FAST
c. Line 733; GXNA
d. Line 744;CLARANS and line 748; FCLARANS
e. Line 761 and 763; DisGeNET
f. Line 780;maTE
g. Line 796;AUC
5. It is clear that authors have used square brackets i.e., () to describe the full form of an abbreviation at various places throughout the manuscript. However, at several places the use of square brackets are confusing. For example, at Line 108; features (mRNAs): I understand mRNA is a feature here. In this case, using features (i.e., mRNAs) instead of features (mRNAs) would be more clear. However, using i.e., is not suitable for Line 204; dependency (similarity). Therefore, authors needs to fix this accordingly throughout the manuscript. A few examples that needs to fix are given below:
a. Line 37 and line 881; optimization-based (fuzzy or evolutionary)
b. Line 70 and line 137; samples (observations)
c. Line 111; features (taxa)
d. Line 204; dependency (similarity)
e. Line 349; nodes (features) and impurity (Gini impurity)
f. Line 369; criteria (dependency, information, distance and consistency [34])
g. Line 373-374); methods (mixture model, regression modeling and t-test)
h. Line 455; indirectly (via other features)
i. Line 469 and line 622; attribute (feature)
j. Line 665; score (rank)
k. Line 666, line 722 and line 744; genes (features)
l. Line 673; SVM-RCE (although they call it RCE-SVM in their paper)
m. Line 755; original discriminative scores (from IG)
n. Line 789; cluster (pathway)

Reviewer 3 ·

Basic reporting

The manuscript has been significantly improved. However, my few points are still not clearly addressed. Consider my review points 4,6, 7, and 15.

Experimental design

Need minor revision

Validity of the findings

Need minor revision

---

## Round 0.3 · accepted · Accept

The authors have revised the manuscript successfully by addressing all the reviewers’ comments.

Reviewer 3 ·

Basic reporting

Manuscript has been revised successfully. It may be accepted in the current form.

Experimental design

Manuscript has been revised successfully. It may be accepted in the current form.

Validity of the findings

Manuscript has been revised successfully. It may be accepted in the current form.